# Inducing the Degradation of Disease-Related Proteins Using Heterobifunctional Molecules

**DOI:** 10.3390/molecules24183272

**Published:** 2019-09-08

**Authors:** Alexandré Delport, Raymond Hewer

**Affiliations:** Discipline of Biochemistry, School of Life Sciences, University of KwaZulu-Natal, Pietermaritzburg 3201, South Africa

**Keywords:** PROTACs, SNIPERs, targeted protein degradation, drug development

## Abstract

Current drug development strategies that target either enzymatic or receptor proteins for which specific small molecule ligands can be designed for modulation, result in a large portion of the proteome being overlooked as undruggable. The recruitment of natural degradation cascades for targeted protein removal using heterobifunctional molecules (or degraders) provides a likely avenue to expand the druggable proteome. In this review, we discuss the use of this drug development strategy in relation to degradation cascade-recruiting mechanisms and successfully targeted disease-related proteins. Essential characteristics to be considered in degrader design are deliberated upon and future development challenges mentioned.

## 1. Introduction

Proteins within the human body are dynamically regulated at the transcriptional, translational and posttranslational level. Failure in these regulatory systems can result in diseases such as cancers [1] and proteopathies, such as the unnatural accumulation of proteins that contribute to several neurodegenerative diseases [2]. Therapeutic strategies to treat these diseases aim to restore affected proteins back to their natural states. Current strategies mostly target either enzymatic or receptor proteins for which specific small molecule ligands can be designed for modulation [3]. However, this classic paradigm is challenged by the identification of druggable targets [4,5] and so efforts to broaden the druggable proteome have been a recent focus. An avenue exploited for this reason is the recruitment of natural degradation cascades for targeted protein removal using heterobifunctional molecules or degraders.

Of the natural protein degradation cascades, the ubiquitin–proteasome (UP) cascade is the best characterized. The enzymes E1, E2 and E3 ligases collectively mediate the tagging of a target protein with ubiquitin allowing the capped (26S) proteasome to recognize and degrade it [6]. Two types of degraders designed to exploit this cascade are protein-targeting chimeric molecules (PROTACs) and specific and non-genetic IAP-dependent protein erasers (SNIPERs) [7]. These molecules consist of a protein-binding ligand (a warhead), a linker and an E3 ligase-binding ligand (recruiter) which allows for target protein binding by the recruited E3 ligase and subsequent degradation (Figure 1). PROTACs commonly recruit cereblon (the substrate receptor for the cullin-ring E3 ligase 4 complex, CLR4) and the von Hippel–Lindau (VHL) E3 ligase complex, while SNIPERs recruit the anti-apoptotic protein, cIAP1. Cellular proteins can also be degraded in a ubiquitin-independent manner by both the 26S proteasome and the uncapped (20S) proteasome [8]. This pathway has been successfully exploited by linking a hydrophobic tag to a target protein-binding ligand [9,10,11]. When the ligand binds the target, the hydrophobic region mimics the unstructured regions of an unfolded protein thereby promoting ubiquitin-independent degradation (Figure 2A). Lysosomal degradation is responsible for the gradual turnover of cytosolic organelles and proteins [12]. Although most lysosomal degradation pathways are non-selective, chaperone mediated autophagy (CMA) is an exception [12]. Here, target protein degradation is dependent on the recognition of an exposed peptide motif by chaperones which bind and deliver them to the surface of the lysosome for degradation [13]. This selective pathway has been used for the development of peptidic CMA-recruiting degraders (Figure 2B).

The design of bifunctional degraders with the ability to remove a protein of interest, thereby reducing its abundance, holds the potential to abrogate all of its functions. On the other hand, traditional small molecule ligands, such as enzyme inhibitors and receptor antagonists, target a specific function or activity of a protein for a modulating affect (for a detailed comparison between degraders and traditional small molecule ligands see an account by Itoh [14]). Therefore, degraders have unique advantages over traditional, function specific-targeting ligands, which do not influence a targets additional functions, such as an alternative role in protein–protein interactions (PPIs) or protein scaffolding. Furthermore, drugs that block a specific binding site are more susceptible to drug resistance, as has been observed for several cancers [15]. In the context of prostate cancer, the androgen receptor (AR) antagonist enzalutamide can become an agonist in the presence of a single mutation in the receptor—which is characteristic of resistance against this drug [11]. By adding a recruiter to enzalutamide, a degrader was produced with anti-proliferative effect against cell lines resistant to the drug alone [11]. More recent studies have also shown that converting promiscuous inhibitors that target kinases into protein degrading molecules significantly improves specificity and binding affinity [16,17,18] and can selectively target specific p38 MAP kinase isoforms [19]. Therefore, these degraders have unique advantages over conventional small molecule drugs. This technology also provides scope for the selection of recruitment strategies best suited for a specific target protein or disease. The potential to preferentially select a degradation pathway and, beyond that, a recruiting mechanism that predominates in the same organelle as the target [20] could possibly allow for a more efficient drug. Further, the option to select different degradation pathways is advantageous in diseases where a particular pathway is in dysfunction. For example, in diseases where the dysregulation of the UP cascade results in a specific target protein proteopathy (as for Huntingtin, Parkinson and Alzheimer’s disease [21,22]), the recruitment of another degradation pathway to remove the accumulated protein may offer an alternative treatment strategy.

In this review, we aim to discuss these bifunctional degraders in their ability to recruit different natural degradation cascades to induce the removal of a selection of disease-related proteins. We specifically focus on the efficiency of each degrader and characteristics needed to be considered in degrader design. These degraders, with their ability to target many protein targets have the potential to expand the druggable proteome and treat numerous diseases, highlighting their importance for future medicinal applications.

## 2. Cereblon-Recruiting PROTACs

Immunomodulatory agents (phthalimide derivatives: lenalidomide, thalidomide and pomalidomide) are a family of ligands found to induce the degradation of transcription factors IKAROS family zinc finger 1 and 3 (IKZF1/3) by interacting with cereblon, triggering their ubiquitination and degradation [23]. Most recently, a phthalimide analogue named TD-106 was developed as a novel IKZF1/3 degrader [24]. Subsequently, these immunomodulatory agents have been exploited to develop cereblon-recruiting PROTACs by linking them to protein-targeting ligands. The resulting bifunctional molecule, as conceptualized in Figure 1A, is designed to specifically degrade targeted proteins through a ubiquitin-dependent pathway. Several groups have used this approach to create bromodomain and extra-terminal motif (BET) protein BRD4 degraders. BET inhibitors have significant anti-proliferative effects on several cancer types suspected to be due to the downregulation of oncogenes [25]. The success seen in numerous preclinical trials as well as in early clinical trials against leukemia, lymphoma and NUT (nuclear carcinoma of the testis) carcinoma shows favorable development of BET targeting molecules as novel anticancer agent [21]. Souza and colleagues [26] linked thalidomide to BET protein inhibitor JQ1 to produce dBET1. This PROTAC yielded a degradation concentration (DC_50_: the concentration at which 50% of the target protein has been degraded) of 460 nM for BRD4 and greater anti-proliferative effect (EC_50_ = 140 nM) when compared to the inhibitor alone (EC_50_ = 1100 nM), as measured by cellular ATP content in the MV4-11 cell line (Table 1). Similarly, Kim, et al. [24] produced TD-428 (TD-106 linked to JQ1) with a potent DC_50_ of 0.32 nM for BRD4 and cytotoxicity concentration (CC_50_) of 20.1 nM in the 22Rv1 prostate cancer cell line. The BET inhibitor BETi-211 was attached to thalidomide to produce dBET246 (DC_50_ < 10 nM, Table 1) with > 50-fold more potent inhibition of growth in nine triple-negative breast cancer cell lines when compared to its inhibitor counterpart, BETi 211 [27]. Crews, Coleman and coworkers [28] made ARV-825 by linking pomalidomide to a triazolo-diazepine acetamide BET protein-binding motif (DC_50_ = 1 nM, Table 1) which caused a depletion of c-Myc and increased apoptosis in a secondary AML cell line and an MCL cell line [28,29,30]. The identification of the azacarbazole-based BET protein inhibitors in 2015 also prompted the development of an additional group of BRD4 degraders [31]. Reported within this family of PROTACs were compounds 21 and 23 which represent the most potent BRD4 degraders developed to date with DC_50_ values of 37 and 51 pM, respectively (Table 1). Compound 23 was further shown to reduce tumor size within a mouse xenograft model where levels of BRD2, 3, 4 and c-Myc were reduced. Taken together, the BET-protein degraders all showed advantages over BET protein inhibitors alone and had significant anti-proliferative effects on several cancer cell lines as well as in an in vivo tumor model.

Research pursuing the development of PROTACs that recruit cereblon has had success in inducing the degradation of another bromodomain-containing protein, BRD9 [32]. dBRD9 was produced using BRD9 inhibitor BI7273 conjugated to lenalidomide with a DC_50_ = 104 nM (Table 1) and limited off-target binding across the BET family of proteins. Further, dBRD9 had significant anti-proliferative effect in an AML cell line. The inability of BET protein inhibitors to distinguish between protein isoforms while degraders show specificity as well as a significant increase in anti-proliferative effect demonstrates the therapeutic potential of these molecules in the treatment of some cancers. It was, however, noted that the dBRD9 PROTAC retained the ability to degrade IKZF, the natural lenalidomide substrate. The degradation of PCAF and GCN5 bromodomain-containing proteins was also successful (with GSK983, DC_50_ = 1.5 nM and 3 nM for PCAF and GCNF bromodomain respectively, Table 1) and represents the first use of PROTACs with anti-inflammatory applications [33].

Kinase inhibitors are currently used for the treatment of numerous cancers and therefore kinase degraders could potentially be used with similar effect. As detailed in Table 1, cereblon-recruiting PROTACs have been shown to effectively degrade several kinases including c-Abl and Bcr-Abl (with a dasatinib-pomalidomide conjugate: DAS-6-2-2-6-CRBN, DC_50_ = 25 nM) [34], CDK2 and CDK9 (with a TAE684-pomalidomide conjugate: TL12-186, DC_50_ = 73 nM and 55 nM) [18], p38δ (with a foretinib-thalidomide conjugate: DC_50_ = 27 nM) [16] and BTK (with a ibrutinib derivative-pomalidomide conjugate: MT802, DC_50_ = 27 nM) [35]. Interestingly, foretinib, a promiscuous kinase inhibitor known to bind 133 kinases, was linked to thalidomide and was shown to degrade only 14 kinases [16]. The increased specificity of the promiscuous degrader, as compared to the inhibitor alone, suggested that a stable ternary complex and PPIs are essential for establishing a functioning PROTAC, which was similarly indicated by Huang, et al. [18]. The development of degraders that are dependent on stability and PPIs is advantageous, since ligands that show weak affinity or are non-inhibitory can still be used [17,36]. Moreover, this allows for the rational design and modification of PROTACs in order to enhance potency and specificity [19,36].

Based on cereblon-recruiting PROTACs, Heightman and his team at Astex pharmaceuticals [37] developed CLIPTACs (in-cell click-formed proteolysis targeting chimeras) which are two separate ligands (one for protein binding and one for cereblon binding) that are able to undergo cycloaddition allowing for the formation of the degrading molecule within the cell (Figure 1B). This technology was used to degrade BRD4 and extracellular signal-regulated protein kinases 1 and 2. The epigenetic erasers sirtuin 2 and histone deacetylases 6 were shown to be degraded in a cereblon dependent manner with SirReals-thalidomide [38] and vorinostat-pomalidomide (DC_50_ = 34 nM) [39] degraders, respectively. Pirin protein was also effectively degraded by recruiting cereblon, using a novel cell-based screening technique, validating the use of PROTACs for targets previously considered undruggable [40].

The success of cereblon-recruiting PROTACs showcases the capacity of CLR4 to degrade a range of disease-related proteins. Currently, most proteins degraded are those related to cancer, particularly BRD4. It would be interesting to see if this recruiting technology could be extended to target other cancer-related proteins such as AR or ER. Further, application of this technology to remove neurodegenerative disease-related proteins, such as mutant huntingtin (mHtt) or tau protein, would also be useful. It should, however, be considered that these molecules may retain their immunomodulatory abilities, potentially posing a challenge in future PROTAC drug development. Nonetheless, the cereblon-recruiting strategy has produced the most efficient degrader developed (pM range) to date.

## 3. VHL-Recruiting PROTACs

The VHL E3 ligase complex is commonly recruited for targeted UP protein degradation (Figure 1A). Initially, these PROTACs were designed as bifunctional peptides using a pentapeptide motif (ALAPYIP) for VHL recruitment and were shown to degrade ER, AR, ER alpha (ERα), FK506 binding protein (reviewed by Raina and Crews [41]), aryl hydrocarbon receptor [42], Akt protein kinase [43] and tau protein [44]. These peptide-based PROTACs lack potency (DC_50_ values between 3–50 µM) to be drug-like, but do represent a novel tool to study protein knock-down. All small molecule PROTACs that recruit VHL were therefore synthesized. Similar to cereblon-recruiting PROTACs, MZ1 and ARV-771 were developed to degrade BRD4. Although MZ1 lacked potency (DC_50_ = 1 µM) when compared to other BRD4 PROTACs, it was shown to have specificity for BRD4, over BRD2 and 3 isoforms [45]. ARV-771 degraded BRD4 with a DC_50_ of 5 nM (Table 1) and resulted in the sustained depletion of c-Myc with an EC_50_ of 1 nM [46], similar to the pomalidomide conjugate equivalent (ARV-825). This PROTAC also reduced leukemia burden in an AML in vivo xenograft model [29] and induced a higher survival rate in a MCL ibrutinib-resistant in vivo xenograft model, when compared to inhibitor alone [30]. MZ1 was used to determine the structure of the VHL-PROTAC-BRD4 ternary complex [36], which displayed specific PPIs between VHL and BRD4 with degradability dependent on linker length. Using the derived model, AT1 was rationally designed with higher specificity for BRD4. The improved specificity that PROTACs show for BRD4, unlike traditional BET protein inhibitors which show comparable affinity for BRD2, 3 and 4, highlights the therapeutic application that BRD4-PROTACs may have in the treatment of cancer, such as MM and AML, for which this isoform is specifically implicated.

The targeting of bromodomain containing protein BRD9 and BRD7 for degradation via the recruitment of VHL has also been successful. The development of the most optimal degrader VZ185 (BRD9; DC_50_ = 4 nM and BRD7; DC_50_ = 34 nM) using two rounds of systematically varying conjugation and linker length allowed for a rational approach to PROTAC design from a previously incompatible recruiting mechanism. This BRD9 VHL-recruiting PROTAC was significantly more potent compared to the cereblon-recruiting dBRD9 [47]. Further, the bromodomain containing chromatin remodeling ATPase complexes SMARCA2 and 4, have also been successfully degraded using the VHL recruiting PROTAC ACBI1. The design of ACBI1 employed co-crystallization techniques to elucidate a high-binding, ternary complex-forming PROTAC which degraded the targets SMARCA2 (DC_50_ = 6 nM), SMARCA4 (DC_50_ = 11 nM) and PBRM1 (DC_50_ = 32 nM) (Table 1). The degrader was subsequently shown to have anti-proliferative effect in several cancer cell lines [48].

Numerous kinases have been implicated in cancer progression as well as several other diseases and present an attractive drug target. However, due to similarities across kinase isoforms and families, the requisite specificity of kinase inhibitors is severely lacking. As summarized in Table 1, VHL-recruiting PROTACs have been designed to degrade kinases and pseudokinases including RIPK2 (with PROTAC_RIPK2, DC_50_ = 1.4 nM) [49], c-Abl (with DAS-2-6-6-2-VHL, DC_50_ = 1 µM) [34], p38α (with VHL PROTAC 1, DC_50_ = 210 nM [16] and SJFα, DC_50_ = 7.2 nM) [19]), p38δ (with SJFδ, 46.2 nM) [19], EGFR (with VHL gefitinib, DC_50_ = 11.7 nM), HER2, (with VHL-lapatinib, DC_50_ = 102 nM), c-Met (with VHL-foretinib, DC_50_ = 66.7 nM) [17] and Fak (with PROTAC-3, DC_50_ = 3 nM) [50]. The lack of degradation of Bcr-Abl and the low potency for c-Abl by VHL-recruiting PROTACs when compared to cereblon-recruiting PROTACs emphasizes the significance of the recruiting ligand for PROTAC design [34]. The development of a multi kinase degrader by linking foretinib to a VHL-recruiting ligand (VHL-PROTAC 1) with the ability to degrade 9 kinases—5 less than its cereblon-recruiting counterpart (CRBN-PROTAC 2)—further highlights the importance of this parameter [16]. These two PROTACs showed specificity for different p38 MAP kinase isoforms, with VHL-PROTAC 1 degrading p38α most efficiently and CRBN-PROTAC 2, p38δ. Recently, the assessment of linker length, specificity and degradability was evaluated using the foretinib-VHL ligand PROTAC, where the preference of a p38 isoform was highly dependent on linker length and the successful development of specific p38α and p38δ degraders shown [18]. The ability to distinguish between p38 MAP kinase isoforms is advantageous since each have distinct functions in the kinase signaling cascades [51]. The development of VHL-recruiting PROTACs with different warheads (kinase inhibitors: lapatinib, gefitinib and afatinib) also affects degradability. While VHL-gefitinib degraded EGFR with highest efficiency, all three were able to remove it, while only VHL-lapatinib was able to degrade HER2 [17] (Table 1). It is therefore clear, that E3 ligase recruiter, linker length and protein ligand selection determine the successful formation of a stable ternary complex and specific PPIs essential for effective degradation and specificity. As before, the PROTACs demonstrated marked improvement in efficacy over the inhibitor alone, as was seen for PROTAC-3 (VHL-defactinib) which showed superiority over clinical candidate defactinib (Fak inhibitor only) in Fak inactivation and impediment of Fak mediate prostate tumor cell line (PC3) invasion [50]. The advantages that PROTACs offer over kinase inhibitors further emphasizes their therapeutic application.

A VHL-recruiting PROTAC has been shown to degrade estrogen-related receptor alpha (ERRα) (PROTAC_ERRα, DC_50_ = 100 nM) [49] with implications in breast cancer treatment. Both an estrogen receptor alpha (ERα) PROTAC (ARV-471, DC_50_ = 2 nM) [52] and androgen receptor targeting PROTACs (AR PROTAC, DC_50_ < 1 nM [53] and ARV-110 [54]) have been developed, although their structures are nondisclosed (Table 1). The ERα-PROTAC was assessed for degradation in a proteomic study of 7600 proteins which concluded that it specifically caused the depletion of ERα and several genes associated with downstream ER cascades, showcasing the specificity of PROTACs [55]. ARV-471 and ARV-110 are both orally bioavailable PROTACs (as tested in an in vivo murine model) [52,53] and have received approval for phase I clinical trials for breast and prostate cancer, respectively [54].

The variability in VHL-PROTAC design has enabled these molecules to degrade a vast number of disease-related proteins, particularly those involved in cancer. VHL-recruiting PROTACs showed different specificity and efficiency for different target proteins when compared to cereblon-recruiting PROTAC counterparts highlighting the importance of recruiting ligand, protein ligand and linker length selection to ensure PPI and stable ternary complex formation essential for degradation. Recruiting VHL does not, seemingly, result in off-target degradation as observed for pthalidimide derivatives. The success of a VHL-recruiting peptidic PROTAC that degraded tau protein, implicated in Alzheimer’s disease [40], suggests that PROTACs can be used to target neurodegenerative related proteins.

## 4. cIAP1 Recruiting SNIPERs

The anti-apoptotic protein, cIAP1, is known to have autoubiquitination abilities via the use of its integrated E3 ligase-like ring finger domain. It has further been shown to be overexpressed in several cancers and represents a possible drug target. Bestatin, a commonly used aminopeptidase inhibitor, has immunomodulatory activity and has been used for targeted depletion of cIAP1 by binding and inducing its autoubiquitination for the treatment of cancer. Natio and coworkers [56] further showed that a bestatin homologue ME-BS was able to induce cIAP1 degradation by autoubiquitination as well. Bestatin, ME-BS, BE04 (ME-BS but ester substituted with an amine), LCL161 and MV1 (IAP antagonists) have subsequently been used to develop SNIPERs by recruiting cIAP1 (Figure 1A). SNIPERs degrade cellular retinoic acid binding protein I (CRABP-I) (with ME-BS conjugated to all-trans retinoic acid, DC_50_ = ~10 µM) [57], CRABP II (with BE04-all-trans retinoic acid) [58], retinoic acid receptor (with BE04-Ch55), ER (with BE04-estrone), AR (with BE04-DHL) [59], ERα (with bestatin-tamoxifen, DC_50_ = ~10 µM [60] and SNIPER(ERα)-87, DC_50_ = 9.6–15.6 nM) [61], transforming acidic coiled-coil-3 (TACC3, with SNIPER(TACC3), DC_50_ = ~10 µM) [62], Bcr-Abl/Abl (with SNIPER(ABL)-38, DC_50_ = ~30 nM), BRD4 (with SNIPER(BRD4)-1), DC_50_ = ~10 nM), PDE4 (with SNIPER(PDE4)-1, DC_50_ = ~10 nM) [61] (Table 1) and mHtt protein (with BE04/MV1-phenyldiazenyl benzothiazole derivative PDB [63,64]). These degraders have therefore been successfully used in the degradation of several cancer and neurodegenerative associated proteins, with arguably more variation compared to cereblon- or VHL-recruiting PROTACs.

The SNIPER induced degradation of CRABP-I reduced neuroblastoma cell migration by 75% [57]. Furthermore, those SNIPERs that target ERα were shown to suppress the growth of tumor cells (IC_50_ = 15.6 nM and 9.6 nM in MCF7 and T47D, respectively) and attenuate tumor proliferation in in vivo murine breast cancer xenograft models [61]. The induced degradation of TACC3 when treated with ≥10 µM SNIPER(TACC3) resulted in apoptosis of both the fibrosarcoma (HT1080) and breast cancer (MCF7) cell lines and cell death was cancer cell specific, not affecting normal cell lines (TIG1 and MRC5) [62]. Uniquely, SNIPERs can simultaneously degrade cIAP1 (autoubiquitination) (Figure 1A) and the target protein which could potentially be advantageous when considering ternary complex formation and the importance of target protein, degrader and E3 ligase concentration ratio. However, this can also be circumvented by using BE04 where the ester of ME-BS is replaced with an amine group [58]. Experimental studies in which degradability of a SNIPER was analyzed by varying the linker length showed similarities with PROTACs, in that linker length altered the efficacy of a degrader [60,64]. This again suggests that a stable ternary complex and PPIs are required for successful protein removal through this recruiting mechanism, although further research into the three-dimensional structure of a SNIPER induced ternary complex is still required.

SNIPERs have therefore been used to degrade several cancer-associated proteins and with similar potency to VHL-recruiting PROTACs (DC_50_ nM range). This degrading technology was also exploited to degrade mHtt protein with direct implication for the neurodegenerative disease [63,64]. Although this SNIPER (MV1-PDB) used a ligand known to bind protein oligomers, rather than the recognition of a specific protein, its design could be further extended to other proteopathies, such as the oligomerization of Aβ peptide, with significance in Alzheimer’s disease. SNIPERs further showcase the importance of linker length in degrader design reinforcing what has been previously shown, that degradability is not solely dependent on the affinity of the ligand for the target.

## 5. Other E3 Ligase Recruiting Protacs

E3 ligases, with their great diversity, have the potential to target a large variety of proteins for degradation. Using bifunctional peptidic PROTACs, CHIP [65], MARCH5, NEDD4L, parkin, SIAH1, βTrCP [66] and Keap 1 [67] E3 ligases have all been recruited and successfully induced the degradation of their target protein. Further, in the development of ligand-based degraders, MDM2 (mouse double minute 2 homologue) is another E3 ligase that has been recruited for the degradation of AR (with nutlin-SARM) [68] and BRD4 (with A1874, DC_50_ = 32 nM) [69]. A1874 (nutlin-JQ1) was significantly more potent than dBETi albeit less efficient than TD-428 and the other BRD4 targeting cereblon- and VHL-recruiting PROTACs that use different inhibitor-based degraders (Table 1). Recently, using small electrophilic ligands linked to known inhibitors of the predominantly nuclear proteins FKBP12 (SLF) and BRD4 (JQ1), a chemical proteomic approach identified the recruitment of DCAF6 and DTL (nuclear CUL4-DDB1 E3 ligase substrates) for targeted protein reduction [20]. The use of electrophilic ligands that can bind a broad range of targets and the subsequent identification of organelle specific E3 ligase substrates [20], highlights a possible advantage of organelle specific E3 ligase selection for specific organelle localized protein targets. Nimbolide’s anti-proliferative effect against breast cancer cells was recently shown to be due to RNF114 E3 ligase binding. Therefore, it was linked to inhibitor JQ1 and used for BRD4 targeted protein degradation by recruiting RNF114 [70]. MDM2, DCAF6, DTL and RNF114 all represent additional recruiters for novel degrader design. Since all small molecule PROTACs and SNIPERs have shown that degradability is dependent on a stable ternary complex and specific PPIs [32,58], the use of different E3 ligases for different target proteins becomes essential. Additionally, as degraders enter the clinic, the E3 ligase levels within specific organelles and tissues may become an important consideration for degrader design [71]. It is, therefore, crucial for future degrader development to focus on the identification of novel specific E3 ligase binding ligands, to broaden recruiter choice for optimal drug design, as has been done recently [20,70].

## 6. Ubiquitin Independent Small Hydrophobic Tag Degraders

As illustrated in Figure 2A, small hydrophobic tag degraders bind to the surface of a target protein and mimic a partially denatured state therefore activating the cells quality control machinery and inducing proteasomal degradation [72]. Two hydrophobic tags have been developed for this purpose; tert-butyl carbamate-protected arginine (Boc3Arg) and an adamantyl group. These proteasome 20S-dependent degraders induce degradation independent of ubiquitin [73]. Boc3Arg, in particular, is suspected to bind directly to the 20S proteasome [73]. This method has been shown to degrade glutathione-s-transferase (with Boc3Arg-nitrobenzoxadiazolyl inhibitor), dihydrofolate reductase (with Boc3Arg trimethoprim) [9], AR receptor (with SARD279, DC_50_ = 1 µM and SARD033, DC_50_ = 2 µM) [11] and HER3 (with TX2-121-1: adamantly-TX1-85-1) [10]. In particular, SARD279 suppressed proliferation with similar efficacy to MDV3100 inhibitor alone in the prostate cancer cell line LNCaP and notably, under MDV3100 resistant conditions as well [11]. The efficiency of AR receptor degradation by SARD279 and SARD033 was linker length dependent [11], suggesting that this component of hydrophobic tag degrader design was also important, as was determined for UP recruiting degraders. Subsequently, the development of novel bifunctional selective estrogen receptor degraders (SERDs) by linking an ERα ligand to various hydrophobic tags has been done. Ligands linked to adamantane groups outperformed those linked to Boc-amino acid groups for ERα downregulation [74]. Hydrophobic tagging has also been shown to degrade tau protein with implications in Alzheimer’s disease [75] and TAR DNA binding protein 43 with implications in amyotrophic lateral sclerosis [76]; this was, however, achieved with peptide-based conjugates. Owing to the large molecular weight of PROTACs and SNIPERs [18], cellular uptake is reduced [37] and the development of degraders that are more drug-like is required. This may be possible using hydrophobic tags, however, these degraders do lack the in vitro potency of PROTACs and SNIPER*s*. Further optimization and development of this form of degrader is required to assess their true potential.

## 7. Chaperone Mediated Autophagy-Recruiting Degraders

The CMA pathway of lysosomal degradation is a type of autophagy that is dependent on the presence of specific pentapeptide motifs within target proteins [13]. These motifs are recognized by chaperones which deliver target proteins to the surface of the lysosome for degradation. These pentapeptide motifs are reminiscent of the peptide motifs used in the development of early peptidic PROTACs [77] and were used in a similar design for peptidic CMA-recruiting degraders (Figure 2B). By conjugating the polyglutamine binding peptide 1, known to bind mHtt protein, and two recruiting pentapeptides, KFERQ (from RNase A) and VKKDQ (from α synuclein), a peptidic degrader was developed with the ability to degrade huntingtin protein [78]. The degrader was able to reduce mHtt protein, in vitro by 42% and improve disease phenotype in two in vivo mouse models [78]. A similar strategy was used to successfully design peptidic CMA degraders that target DAPK1, α-synuclein and PSD-95 [79]. The DAPK1 targeting degrader displayed neuroprotectivity in an in vivo mouse stroke model [79]. Although a ligand-based CMA degrader has not been developed, the similarities with early peptidic PROTACs and their success in vivo demonstrates the capability of this recruiting strategy. In order to increase the efficacy of this recruiting mechanism as a therapeutic strategy, an all ligand-based degrader is necessary. Several inhibitors that bind cellular chaperones such as the Hsp70 family have been identified [80]. Further, using PDB known to bind soluble oligomers such as mHtt protein, a ligand-based CMA degrader could be synthesized similarly to the huntingtin protein-targeting SNIPER. The successful recruitment of this pathway would represent a novel class of protein degrader and potentially further expand the druggable proteome.

## 8. Conclusions

Current drug development strategies that use small molecules to target specific protein activities result in a large portion of the proteome being overlooked as undruggable. The generation of degraders that specifically remove disease-related proteins by recruiting native degradation pathways is a novel drug development strategy with the potential to expand the druggable proteome. This review specifically outlined those degradation cascades recruited for the removal of disease-related proteins. The different recruiting strategies discussed are either ubiquitin dependent or independent and each has unique advantages and disadvantages (Table 2). The development of both the ubiquitin dependent degraders, PROTACs and SNIPERs, have shown most success. Together, as ligand-based degraders, they have induced the degradation of at least 30 different cancer-, neurodegenerative- and inflammatory disease-related proteins. This number increases to a minimum of 36 if UP recruiting peptidic degraders are included. Further, both ubiquitin independent degradation and the CMA cascade have also successfully been recruited for targeted protein degradation. Therefore, a total current estimate of 43 different proteins have been targeted for removal using this technology, showcasing the potential to treat a variety of human diseases.

For successful degradation of a target protein, several groups have concluded that a stable ternary complex and identifiable PPIs between recruiter and target protein are required, meaning that degraders can be rationally designed. Although initially the degrader design was empirical, increased effort towards a more rational approach has been observed in the past three years. In general, this involves co-crystallization of the ternary complex and subsequent systematic in silico improvement of the degraders induced target protein–recruiter interactions [19,36,47,48]. This technique has successfully been used to develop a degrader for a target protein–recruiter complex (BRD9-VHL), previously deemed non-PROTACable [47]. Currently, partial PROTACs, which consist of a VHL- or cereblon-recruiting ligand with simplistic warhead attachment and varying linker length, are commercially available and allow for a larger degrader screening library, and therefore, an increased probability of identifying a ‘hit’ compound. Since these specific interactions are more dependent on linker length than ligand binding affinities [81], a large repertoire of already identified small molecules that lacked potent inhibition or that had low binding affinity, currently exists, that can be repurposed into a degrader.

Although PROTACs have several advantages over small molecule inhibitors and have been shown to target a wide range of disease-associated proteins, their development into clinically-relevant drugs will face several challenges due to their large molecular weight, low absorption, low bioavailability and general non-drug like properties [71]. Some success in circumventing these problems has already been shown with the development of CLIPTACs [37], while orally bioavailable PROTACs do exist [52,54]. Additionally, the first degraders have been approved for phase I clinical trials for the treatment of breast and prostate cancer [48]. Another consideration is the resynthesis rate of target protein versus recruited E3 ligase [71]. A study to determine the tissue distribution of transmembrane E3 ligases analyzed the total RNA of 22 normal human tissue samples and found that a majority were ubiquitously expressed [82]. However, several were tissue specific such as RNF183 expressed predominantly in the kidney and testis, RHF182 and 175 in the brain and spinal cord and RHF186 in the brain, pancreases, intestines and kidneys [82], indicating that E3 ligases give tissue-dependent expression profiles, and therefore, possibly dependent resynthesis rates. If this is the case, the possibility exists where either the target or the E3 ligase will form a rate limiting component or both, in which case higher degrader concentration in a specific disease targeted tissue could lead to binary rather than ternary complex formation and an ineffective drug (the so-called hook effect). Therefore, careful consideration of the recruited E3 ligase and the tissue in which the drug is expected to act is essential and the need for additional E3 ligase recruiting ligands is emphasized. Further some E3 ligases are organelle specific [83,84], therefore it may also be beneficial to select an E3 ligase that predominates in the same organelle as the target, especially if the level and resynthesis rate of the organelle localized E3 ligase is higher than an E3 ligase ubiquitously expressed throughout the cell. It would be of significant interest to determine the protein level of those E3 ligases used in degrader design to develop a tissue and organelle profile for subsequent ease in recruiter selection. PROTACs have shown significant promise as a novel class of drug, with potency in the pM range [31], success in numerous in vivo studies [29,30,61,78,79] and the first orally bioavailable degraders starting phase I clinical trials [54]. Future PROTAC development must, however, focus on addressing their unique challenges to ensure further success. The use of ubiquitin independent and CMA degradation has also been highlighted and may represent a novel treatment strategy for diseases in which protein removal via the UP cascade has become dysregulated (e.g., neurodegenerative diseases). However, these degraders require further optimization and development before their true potential as therapeutic candidates can be realized.

## Figures and Tables

**Figure 1 molecules-24-03272-f001:**
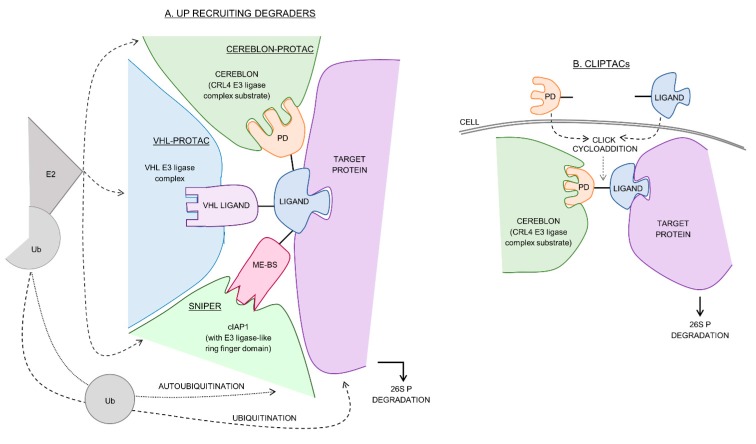
The schematic representation of the different ubiquitin dependent degraders. (**A**) Protein-targeting chimeric molecules (PROTACs) and specific and non-genetic IAP-dependent protein erasers (SNIPERs) bind to the target protein and recruit their respective E3 ligase which, in turn, is then bound by a cognate E2 ligase resulting in the ubiquitination (Ub, ubiquitin) of the target protein and 26S proteasomal (P) degradation. cIAP1 can undergo autoubiquitination and be degraded simultaneously with the target protein. (**B**) In-cell click-formed proteolysis targeting chimeras (CLIPTACs) comprise two separate ligands that undergo cycloaddition (click) in the cell forming a bifunctional degrader, subsequently allowing for target protein degradation. PD, phthalimide derivative (lenalidomide, thalidomide, pomalidomide or TD-106), CRL4, cullin-ring E3 ligase 4 complex; VHL, Von Hippel–Lindau E3 ligase complex; cIAP1, anti-apoptotic protein 1; ME-BS, bestatin derivative.

**Figure 2 molecules-24-03272-f002:**
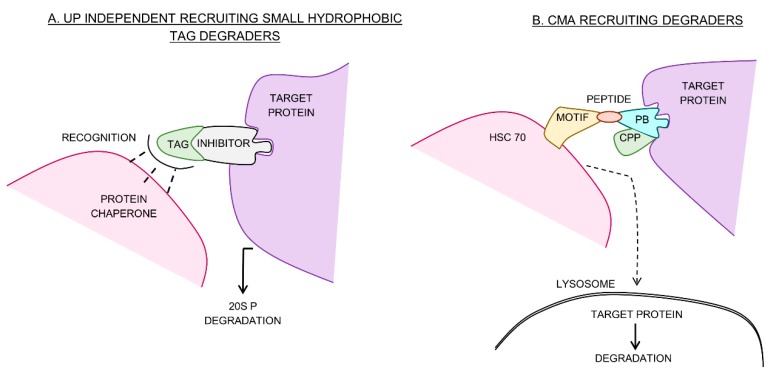
The schematic representation of the different ubiquitin independent degraders. (**A**) Ubiquitin independent small hydrophobic tag degraders, consisting of an inhibitor linked to a hydrophobic tag (TAG: Boc3Arg or an adamantyl group), bind to the target protein where they mimic unfolded regions which, when recognized by protein chaperones, results in 20S proteasomal (P) degradation. (**B**) chaperone mediated autophagy (CMA)-recruiting degraders link Hsc70 with the target protein via a bifunctional peptide that consists of a cell penetrating peptide (CPP), protein binding domain (PB) and Hsc70 recognition motif (MOTIF). Hsc70 then shuttles the target protein to the surface of the lysosome for degradation.

**Table 1 molecules-24-03272-t001:** An overview of the recruiting mechanism-specific degraders in relation to their target protein, disease and degradation efficiency.

Type of Degrader	Degrader Name	Degrader Target	Disease	DC_50_ ^a^ (nM)	Reference
Cereblon-recruiting PROTACs:	dBET1	BRD4	Cancer (MM, AML)	430.0	Winter, et al. [26]
	TD-428	BRD4	Cancer (MM)	32.0 × 10^−2^	Kim, et al. [24]
	ARV-825	BRD4	Cancer (Burkitt lymphoma)	1.0	Lu, et al. [28]
	BETd-246	BRD4	Cancer (breast)	10.0	Bai, et al. [27]
	Compound 21	BRD4	Cancer (MM, AML)	3.7 × 10^−2^	Zhou, et al. [31]
	Compound 23	BRD4		5.1 × 10^−2^	
	dBRD9	BRD9	Cancer (AML)	104.0	Remillard, et al. [32]
	GSK983	PCAF	Anti-inflammatory diseases	1.5	Bassi, et al. [33]
		GCN5		3.0	
	DAS-6-2-2-6-CRBN	cAbl	Cancer (chronic myelogenous leukemia)	25.0	Lai, et al. [34]
		Bcr-Abl			
	TL12-186	CDK2	Cancer, rheumatoid arthritis, and idiopathic pulmonary fibrosis	73.0	Huang, et al. [18]
		CDK9		55.0	
	CRBN-PROTAC 2	p38δ	Cancer and Diabetes	27.0	Bondeson, et al. [16]
	MT-802	BTK	Chronic lymphocytic leukemia	9.1	Buhimschi, et al. [35]
	pomalidomide-Vorinostat	HDAC6	Cancer (AML, ovarian, hepatocellular carcinomas)	32.0	Yang, et al. [39]
VHL-recruiting PROTACs:	MZ1	BRD4	Cancer (NSCLC)	1000.0	Zhong, et al. [65]
	ARV-771	BRD4	Cancer (castration-resistant prostate cancer)	5.0	Raina, et al. [46]
	VZ185	BRD9	Cancer (cervical, NSCLC)	4.0	Zoppi, et al. [47]
		BRD7		34.0	
	ACBI1	SMARCA2	Cancer (AML)	6.0	Farnaby, et al. [48]
		SMARCA4		11.0	
		PBRM1		32.0	
	PROTAC_RIPK2	RIPK2	Auto-inflammatory diseases (Blau syndrome, early-onset sarcoidosis)	1.4	Bondeson, et al. [49]
	PROTAC_ERRα	ERRα	Cancer (breast)	100.0	
	VHL-PROTAC 1	p38α	Cancer	210.0	Bondeson, et al. [16]
	SJFα	p38α		7.2	Smith, et al. [19]
	SJFδ	p38δ	Cancer and diabetes	46.2	
	VHL-lapatinib	EGFR	Cancer (glioblastoma multiforme, NSCLC)	39.2	Burslem, et al. [17]
	VHL-gefitinib			11.7	
	VHL-afatinib			215.8	
	VHL-lapatinib	HER2		102.0	
	VHL-Foretinib	c-MET		66.7	
	PROTAC-3	Fak	Cancer (malignant pleural mesothelioma, ovarian)	3.0	Cromm, et al. [50]
Nondisclosed recruiting PROTACs:	ARV-471	ERα	Cancer (breast)	2.0	Flanagan, et al. [52]
	AR PROTAC	AR	Cancer (prostate)	1.0	Neklesa, et al. [53]
SNIPERs:	SNIPER(CRABP-I)	CRABP-I		~10,000.0 ^b^	Itoh, et al. [57]
	SNIPER(ERα)-87	ERα (MCF-7 cells)	Cancer (breast)	3.0	Ohoka, et al. [61]
		ERα (T47D cells)	Cancer (breast)	9.6	
	SNIPER(TACC3)	TACC3	Cancer (ovarian, breast, squamous cell carcinoma, lymphoma)	~10,000.0 ^b^	Ohoka, et al. [62]
	SNIPER(ABL)-38	cABL/BCR-ABL	Cancer (chronic myelogenous leukemia, MM, AML)	30.0	Ohoka, et al. [61]
	SNIPER(BRD4)-1	BRD4		10.0	
	SNIPER(PDE4)-9	PDE4		10.0 ^b^	
MDM2-recruiting PROTACs:	A1874	BRD4	Cancer (MM, AML)	32.0	Hines, et al. [69]
UP independent HyT degraders:	SARD279	AR	Cancer (prostate)	1000.0	Gustafson, et al. [11]
	SARD033			2000.0	
CMA-recruiting degraders:	TAT-GluN2Bct-PP	DAPK1	Neuroprotectivity (stroke)	~50,000.0 ^b^	Fan, et al. [79]

^a^ DC_50_: the concentration where 50% of the protein has been degraded; ^b^ The DC_50_ value was not given but inferred from experimental results. HyT, Hydrophobic tag. MM, multiple myeloma; AML, acute myeloid leukemia; NSCLC, non-small cell lung cancer.

**Table 2 molecules-24-03272-t002:** The advantages and disadvantages of the different degradation strategies.

Strategy		Advantages	Disadvantages
**Ub Dependent**	Cereblon-recruiting PROTACs	> Increases specificity of promiscuous inhibitors > Most efficient degraders (DC_50_ in pM range)	> Possible off-target degradation (e.g., IKZF)
	CLIPTACs	> More drug-like scaffold	> Small target protein test group
	VHL-recruiting PROTACs	> Increases specificity of promiscuous inhibitors > No off-target degradation	> Lack efficiency when compared to cereblon PROTACs (DC_50_ nM range)
	SNIPERs	> First small molecule degrader to target a neurodegenerative disease-related protein	> Autoubiquitination and simultaneous degradation of cIAP ^a^ > Lack efficiency when compared to cereblon PROTACs (DC_50_ in nM range)
**Ub Independent**	HyT degraders	> More drug-like scaffold	> Lack efficiency (DC_50_ in mM range)
	CMA degraders	> Could potentially be used to treat diseases where the UP cascade is dysfunctional	> Peptidic structure

^a^ Could potentially be advantageous when considering the importance of ternary complex formation and the ratio of target protein, degrader and E3 ligase concentrations.

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
