# Peer review of "Inducing the Degradation of Disease-Related Proteins Using Heterobifunctional Molecules"

_molecules, 2019, doi:10.3390/molecules24183272_

Round 1

Reviewer 1 Report

    This manuscript provides a near-complete review of emerging molecules used for targeted protein degradation. They nicely summarize interesting design considerations with an eye towards recent clinical applications. The authors provide a table to summarize the notable molecules and targets that have been developed and described in recent literature. I felt that one of the most original and important aspects of the review is that the authors highlight the need to consider different E3 ubiquitin ligase recruitment strategies in different tissue contexts. 

    I do not believe the review is ready for publication in its current form. However, with some restructuring and addition of some important missing references, I believe the authors may be able to address these concerns. Below I’ve listed major and minor concerns.

Major:

Structure

 -      In the conclusions section, the authors do a good job of laying out why one would choose one degradation pathway over another. However, this clarifies why this review is important, so I feel that readers would be better served by presenting these ideas in the introduction to motivate the content. I would also like the authors to comment further on the potential issue of tissue specificity of certain degradation strategies.

-      I would like the authors to provide more motivation/significance for the need of the different degradation strategies, including the ubiquitin-independent strategies. Why should readers feel it’s important to consider these different strategies? Is there a clear rational for choosing one degradation pathway versus another?

-      I felt some of the paragraphs unnecessarily duplicate information in tables and reduce readability (for example by listing the inhibitor constants). If possible, the text should be devoted for perspective and analysis rather than data.

Content:

 -      There are other protein degraders that are important for the degradation of disease-related proteins. For example, Fulvestrant is a widely used molecule, and similar selective estrogen receptor degrader (SERD) molecules are discussed in Cermakova et al., Molecules (2018). It would be helpful for the authors to discuss how these molecules fit into the more recent PROTAC concepts they describe, especially since they degrade target proteins in disease settings.

-      Although authors aim to provide an exhaustive list of PROTAC protein targets, there are additional recent targets missed, for example those targeting SWI/SNF complexes: see Farnaby, et al. Nat. Chem. Biol. 15, 672–680, (2019). There are also likely to be many others, so an exhaustive list might be difficult to achieve.

-      Authors should add a figure describing CLIPTACs and some of the examples described in the text.

-      Authors comment on linker length and affinity. How are these parameters optimized for PROTAC development? Are there rational approaches, or is it purely empirical?

-      A small table summarizing the advantages and disadvantages of the different degradation strategies would be valuable.

Minor:

 ·     Figure 1 – the E3 ubiquitin ligases should be labeled as such inside the figure itself given the frequency of describing the roles of the E3 ligases in the text.

 ·     Line 8 – the comma after “strategies” is incorrect in English and should be removed

 ·     Line 9 – “results” should be “result” to match the plural “strategies”

 ·     Line 75 – instead of “current”, “conventional” or “traditional” is more appropriate

 ·     Line 79-80 – conclusion statement is an incomplete sentence

 ·     Line 180 – authors should explain the phrase “therapeutic application of these molecules is clear”

 ·     Line 233 - Introduction of precursors such as bestatin would also improve readability.

 ·     Line 243 – “PDB” is not described in Table 1. Please check that all compounds are listed. 

 ·     Line 295 – sentence is unclear and may contain a grammatical error

 ·     Line 301 – “molecule” should be “molecular”

 ·     Line 335 – add hyphens to increase readability as follows: “cancer-, neurodegenerative-, and inflammatory disease-related proteins.”

 ·     Line 348 – This is a fast-moving field. Instead of providing an exact number of different protein targets, the authors should consider using language such as “at least X” targets.

 ·     Line 362 – “evidenced” is confusing even to a native English speaker. I would suggest “realized” or “proven” as alternatives.

 ·     In many lines, authors incorrectly use “xenograph” instead of the correct “xenograft.”

Author Response

We thank the Reviewers for their valuable time and expert opinion of our manuscript. We have assessed each comment carefully and critically and believe we have made the necessary amendments to the manuscript to the best of our ability. Below, we provide a detailed, point-by-point, response to each comment, indicating how we have addressed the comment and specify the changes made to the manuscript. Each amendment to the manuscript is also clearly indicated through track changes in the attached, revised version of the manuscript. We feel that effecting these changes has improved the quality of the manuscript and we appreciate the value that the Reviewers has imparted on our paper. We hope that these adjustments are suitable and to the satisfaction of the Reviewers. 

Response to Reviewer 1:

Comment 1: The Reviewer wrote " In the conclusions section, the authors do a good job of laying out why one would choose one degradation pathway over another. However, this clarifies why this review is important, so I feel that readers would be better served by presenting these ideas in the introduction to motivate the content. I would also like the authors to comment further on the potential issue of tissue specificity of certain degradation strategies."

Author Reply:

Point 1: We agree with the Reviewer and have added a paragraph within the Introduction that begins on line 83 when Track changes are set to Simple Markup. This paragraph aims to introduce the reader to concept of selecting one degradation system over another and the benefits thereof. The paragraph reads: ‘This technology also provides scope for the selection of recruitment strategies best suited for a specific target protein or disease. The potential to preferentially select a degradation pathway and, beyond that, a recruiting mechanism that predominates in the same organelle as the target [20] could possibly allow for a more efficient drug. Further, the option to select different degradation pathways is advantageous in diseases where a particular pathway is in dysfunction. For example, in diseases where the dysregulation of the UP cascade results in a specific target protein proteopathy (as for Huntingtin, Parkinson and Alzheimer’s disease [21,22]), the recruitment of another degradation pathway to remove the accumulated protein may offer an alternative treatment strategy.’

Point 2: We agree that the potential issue of tissue specificity is an important concept, warranting further commentary within this manuscript. As such, we have expanded on this issue within the Conclusion and have provided more detailed supporting information and analysis. The paragraph added reads: ‘Another consideration is the resynthesis rate of target protein versus recruited E3 ligase [71]. A study to determine the tissue distribution of transmembrane E3 ligases analyzed total RNA of 22 normal human tissue samples and found that a majority were ubiquitously expressed [82]. However, several were tissue specific such as RNF183 expressed predominantly in the kidney and testis, RHF182 and 175 in the brain and spinal cord and RHF186 in the brain, pancreases, intestines and kidneys [82] indicating that E3 ligases give tissue-dependent expression profiles and therefore possibly dependent resynthesis rates. If this is the case, the possibility exists where either the target or the E3 ligase will form a rate limiting component or both, in which case higher degrader concentration in a specific disease targeted tissue could lead to binary rather than ternary complex formation and an ineffective drug (the so-called hook effect). Therefore, careful consideration of the recruited E3 ligase and the tissue in which the drug is expected to act is essential and the need for additional E3 ligase recruiting ligands is emphasized. Further some E3 ligases are organelle specific [83,84], therefore it may also be beneficial to select an E3 ligase that predominates in the same organelle as the target especially if the level and resynthesis rate of the organelle localized E3 ligase is higher than an E3 ligase ubiquitously expressed throughout the cell. It would be of significant interest to determine the protein level of those E3 ligases used in degrader design to develop a tissue and organelle profile for subsequent ease in recruiter selection.’

Comment 2: The Reviewer wrote: ‘I would like the authors to provide more motivation/significance for the need of the different degradation strategies, including the ubiquitin-independent strategies. Why should readers feel it’s important to consider these different strategies? Is there a clear rational for choosing one degradation pathway versus another?’

Author Reply: To address this comment we added a paragraph in the Introduction in which we outline the benefit and the necessity of alternative degradation systems (paragraph beginning line 83 when the Track Changes are set to Simple Markup). Additionally, we refer to ubiquitin pathway dysregulation and propose this as an example of when the use of ubiquitin-independent strategies may be useful. The Reviewer also requested a small table summarising the advantages and disadvantages of the different degradation strategies. This suggestion was highly useful and the requested table was inserted into the manuscript as Table 2. The disadvantages of each system (as per the table) provides implied rational for different degradation systems while the advantages provide considerations and rational for selecting a particular degradation system.  

Comment 3: The Reviewer wrote: ‘I felt some of the paragraphs unnecessarily duplicate information in tables and reduce readability (for example by listing the inhibitor constants). If possible, the text should be devoted for perspective and analysis rather than data.’

Author reply: We thank the Reviewer for this comment. In the process of making the reviewer’s corrections, we have added text to the manuscript. We paid specific attention to adding text that was interpretive and perspective in nature and not data / value based. In doing so, we aimed to reduce the overall concentration of sentences laden with data points and values. We also attempted to remove/ reduce the data points/ DC50 values etc. however, when doing so we found that the clarity of distinction between the different degraders is lost and that this would also require additional effort from the reader to flip between the text and the table. As such, we do hope that the readers will benefit from having the DC50 values quantified within the text. We also inserted more information into the table to ensure that is provides additional information to the text and has ‘stand-alone’ value.

Comment 4: The Reviewer wrote: “There are other protein degraders that are important for the degradation of disease-related proteins. For example, Fulvestrant is a widely used molecule, and similar selective estrogen receptor degrader (SERD) molecules are discussed in Cermakova et al., Molecules (2018). It would be helpful for the authors to discuss how these molecules fit into the more recent PROTAC concepts they describe, especially since they degrade target proteins in disease settings.”

Author reply: To address this omission we included a paragraph to the section on ubiquitin independent small hydrophobic tag degraders. Specifically, the sentences added read: ‘Subsequently, the development of novel bifunctional selective estrogen receptor degraders (SERDs) by linking an ERα ligand to various hydrophobic tags has been done. Ligands linked to adamantane groups outperformed those linked to Boc-amino acid groups for ERα downregulation’ and cited the relevant paper by Wang and co-workers, 2018. We also felt it necessary to clarify the title of the paper through the addition of the words ‘…using heterobifunctional molecules’. This title is perhaps more appropriate to the focus of the paper and serves to limit the scope of the paper to designed bifunctional molecules

Comment 5: The Reviewer wrote: ‘Although authors aim to provide an exhaustive list of PROTAC protein targets, there are additional recent targets missed, for example those targeting SWI/SNF complexes: see Farnaby, et al. Nat. Chem. Biol. 15, 672–680, (2019). There are also likely to be many others, so an exhaustive list might be difficult to achieve.’

Author reply: To address this omission we added the Farnaby et al., 2019 reference through the following description in the section under VHL-recruiting PROTACS: ‘Further, the bromodomain containing chromatin remodeling ATPase complexes SMARCA2 and 4, have also been successful degraded using the VHL recruiting PROTAC ACBI1. The design of ACBI1 employed co-crystallization techniques to elucidate a high-binding, ternary complex-forming PROTAC which degraded the targets SMARCA2 (DC50 = 6 nM), SMARCA4 (DC50 = 11 nM) and PBRM1 (DC50 = 32 nM) (Table 1). The degrader was subsequently shown to have anti-proliferative effect in several cancer cell lines [48].’ Under the same section, we also provided reference to the Med Chem 62, 699-726, 2019 paper of Zoppi and co-workers. We have also added the Med Chem paper by Wang et al., 2018 (as discussed above in relation to SERDs) as well as Nat. Chem. Biol. 2019, 15, 737–746 and Nat. Chem. Biol. 2019. 

Comment 6: The Reviewer wrote: ‘Authors should add a figure describing CLIPTACs and some of the examples described in the text.’

Author reply: Agreed. A schematic representation illustrating the structure and mechanism of CLIPTACs was added to Figure 1. Further, analysis of CLIPTACs under provided under the section on cereblon-recruting PROTACS and in Table 2 added in the Conclusion.

Comment 7. The Reviewer wrote: ‘Authors comment on linker length and affinity. How are these parameters optimized for PROTAC development? Are there rational approaches, or is it purely empirical?

Author reply: To address these comments within the manuscript the following paragraph was included: ‘For successful degradation of a target protein several groups have concluded that a stable ternary complex and identifiable PPIs between recruiter and target protein are required, meaning that degraders can be rationally designed. Although initially degrader design was empirical, increased effort towards a more rational approach has been observed in the past three years. In general, this involves co‑crystallization of the ternary complex and subsequent systematic in silico improvement of the degraders induced target protein-recruiter interactions [19, 36, 47, 48]. This technique has successfully been used to develop a degrader for a target protein-recruiter complex (BRD9-VHL), previously deemed non‑PROTACable [47]. Currently, partial PROTACs, which consist of a VHL- or cereblon-recruiting ligand with simplistic warhead attachment and varying linker length, are commercially available and allow for a larger degrader screening library and therefore an increased probability of identifying a HIT compound. Since these specific interactions are more dependent on linker length than ligand binding affinities [81] a large repertoire of already identified small molecules that lacked potent inhibition or had low binding affinity exists that can be repurposed into a degrader.’

Comment 8: The Reviewer wrote: ‘A small table summarizing the advantages and disadvantages of the different degradation strategies would be valuable.’

Author reply: We agree with this and feel that the addition of Table 2 which summarizes this information has added value to the manuscript. This table is introduced in line 373 of the revised manuscript (when viewed under Simple Markup) “The different recruiting strategy discussed are either ubiquitin dependent or independent and each has unique advantages and disadvantages (Table 2).’ 

Minor Comments. We have endeavored to implement all minor corrections suggested by the Reviewer to the best of our ability. In addition to the comments made and addressed below, we have also re-evaluated our manuscript and sought to improve the grammar throughout.   

Comment 9: The Reviewer wrote: ‘Figure 1 – the E3 ubiquitin ligases should be labeled as such inside the figure itself given the frequency of describing the roles of the E3 ligases in the text.’

Author reply: The E3 ubiquitin ligases have been labelled inside the figure as suggested.

Comment 10: The Reviewer wrote: ‘Line 8 – the comma after “strategies” is incorrect in English and should be removed.’

Author reply: Line 8 (now line 9): The comma after strategies was removed.

Comment 11: The Reviewer wrote: ‘Line 9 – “results” should be “result” to match the plural “strategies”.’

Author reply: Line 9 (now line 10): ‘results’ has been changed to ‘result’

Comment 12: The Reviewer wrote: ‘Line 75 – instead of “current”, “conventional” or “traditional” is more appropriate’

Author reply: The word ‘current’ has been changed to ‘conventional’ as suggested.

Comment 13: The Reviewer wrote: ‘Line 79-80 – conclusion statement is an incomplete sentence.’

Author reply: This sentence has been corrected and Line 79-80 (now line 94 - 96) now reads: “These degraders, with their ability to target many protein targets have the potential to expand the druggable proteome and treat numerous diseases, highlighting their importance for future medicinal applications”

Comment 14: The Reviewer wrote: ‘Line 180 – authors should explain the phrase “therapeutic application of these molecules is clear”

Author reply: Line 180 (now line 192) was re-written to clarify the meaning of the sentence. The sentence now reads “The improved specificity that PROTACs show for BRD4, unlike traditional BET protein inhibitors which show comparable affinity for BRD2, 3 and 4, highlights the therapeutic application BRD4 PROTACs may have in the treatment of cancer, such as MM and AML, for which this isoform is specifically implicated’

Comment 15: The Reviewer wrote: ‘Line 233 - Introduction of precursors such as bestatin would also improve readability.’

Author reply: The following description was added (starting line 258) as a means to introduce bestatin: ‘Bestatin, a commonly used aminopeptidase inhibitor, has immunomodulatory activity and has been used for targeted depletion of cIAP1 by binding and inducing it’s autoubiquitination for treatment of cancer. Natio and coworkers [56] further showed that a bestatin homologue ME-BS was able to induce cIAP1 degradation by autoubiquitination as well.’

Comment 16: The Reviewer wrote:Line 243 – “PDB” is not described in Table 1. Please check that all compounds are listed.

Author reply: Only compounds where DC50 values were specified or in some cases could be inferred from results were included in Table 1 (i.e. as a comparison for efficiency/potency). A DC50 for compound PDB was not given nor could be inferred and therefore was not included in Table 1. On the basis of this suggestion, we have now moved the in-text reference to Table 1 such that it precedes the discussion on PDB .

Comment 17:  The Reviewer wrote: ‘Line 295 – sentence is unclear and may contain a grammatical error.’

Author reply: Line 295 (now line 331) has been corrected and now reads: ‘In particular, SARD279 suppressed proliferation with similar efficacy to MDV3100 inhibitor alone in the prostate cancer cell line LNCaP and notably, under MDV3100 resistant conditions as well [11].

Comment 18: The Reviewer wrote: ‘Line 301 – “molecule” should be “molecular”

Author reply: This has been corrected.

Comment 19: The Reviewer wrote: ‘Line 335 – add hyphens to increase readability as follows: “cancer-, neurodegenerative-, and inflammatory disease-related proteins.”

Author reply: Hyphens have been added as indicated to improve readability.

Comment 20:  The Reviewer wrote: ‘Line 348 – This is a fast-moving field. Instead of providing an exact number of different protein targets, the authors should consider using language such as “at least X” targets.’

Author reply: This sentence has been corrected in line with the Reviewers suggestion and we agree that it this style is better suited to a continuously updating field.

Comment 21: The Reviewer wrote: ‘Line 362 – “evidenced” is confusing even to a native English speaker. I would suggest “realized” or “proven” as alternatives.’

Author reply: As suggested, the word ‘evidenced’ has been replaced with ‘realized’.

Comment 22: The Reviewer wrote: ‘In many lines, authors incorrectly use ‘xenograph’ instead of the correct ‘xenograft’

Author reply: Each use of the word ‘xenograph’ has been corrected to ‘xenograft’.

Reviewer 2 Report

Chemical protein knockdown by PROTACs and SNIPERs, which are degraders for their targeted proteins, is one of hot topics in medicinal chemistry field, and has attracted much attention recently. Therefore, reviews on this theme are very interesting and will be useful for many readers. In this review, the authors, focusing on this technology, introduced and discussed reported PROTACS and SNIPERs. This review is basically well-written and includes important information. Accordingly, this paper should be worth of publication after the following revision.

(i) This review is titled as “Inducing the degradation of disease-related proteins.” However, information on diseases of proteins targeted by each PROTAC is not enough. The authors should add it on Table 1 and make more comments on this in the main body.

(ii) Figure 1: PROTACs and SNIPERs are basically small molecules. On the other hand, cereblon, VHL ubiquitin, and other proteins are macromolecules. I mean that the size of the molecules is not accurate to the model. Please revise it.

(iii) Page 2, line 79: “The significance being that these degraders represent a novel therapeutic strategy to potentially treat numerous diseases.” This sentence is strange.

(iv) Page 3, line 81: “Immunomodulatory agents (phthalimide derivatives: lenalidomide, thalidomide and pomalidomide) were the first ligands found to induce protein degradation.” The word “first” is incorrect. Nuclear receptor ligands have long been known to degrade their target protein degradation.

(v) The authors should refer to two recent important papers: Nat. Chem. Biol. 2019, 15, 737–746.; Nat. Chem. Biol. 2019, 15, 747–755.

(vi) Page 2, line 65: “This is advantageous over other modulating drugs that target specific activities without influencing other functions (such as roles in protein-protein interactions, PPIs).” This sentence is misunderstandable. If a targeted protein has roles to link two proteins, the PPI should be affected.

(vii) Page 2, line 67: “drugs that block a specific binding site are more susceptible to drug resistance, as has been observed for several cancers…It was shown that by adding a degradation recruiting agent to enzalutamide this resistance can be overcome.” PROTACs may have the same problem and there are no guarantees they overcome.

 (vii) The authors should make more detailed comments on the difference between traditional PROTACs/SNIPERs and traditional ligands such as receptor antagonist/enzyme inhibitors: between reduce in protein levels and protein functional inhibition. In this point, the authors should refer to a recent review (Chem Rec 2018, 18, 1681-1700.).

Author Response

We thank the Reviewers for their valuable time and expert opinion of our manuscript. We have assessed each comment carefully and critically and believe we have made the necessary amendments to the manuscript to the best of our ability. Below, we provide a detailed, point-by-point, response to each comment, indicating how we have addressed the comment and specify the changes made to the manuscript. Each amendment to the manuscript is also clearly indicated through track changes in the attached, revised version of the manuscript. We feel that effecting these changes has improved the quality of the manuscript and we appreciate the value that the Reviewers has imparted on our paper. We hope that these adjustments are suitable and to the satisfaction of the Reviewers. 

Response to Reviewer 2:

Comment (i): The Reviewer wrote: ‘This review is titled as “Inducing the degradation of disease-related proteins.” However, information on diseases of proteins targeted by each PROTAC is not enough. The authors should add it on Table 1 and make more comments on this in the main body.’

Author reply: Agreed. In order to address this comment we added the disease for which each target was specifically implicated into Table 1. In addition, those diseases specified by the original authors of the degraders were noted in the text. It is acknowledged that there may be other diseases associated with the target proteins; however, these were not included as we feel that extends beyond the scope of this paper.

Comment (ii): The Reviewer wrote: ‘Figure 1: PROTACs and SNIPERs are basically small molecules. On the other hand, cereblon, VHL ubiquitin, and other proteins are macromolecules. I mean that the size of the molecules is not accurate to the model. Please revise it.

Author reply: This is a highly valid point and in order to address it, we re-drew Figure 1. While we did not aim to draw the molecules to size, we ensured a greater distinction in size between the degraders and the proteins to convey a more accurate representation of the interaction.

Comment (iii): The Reviewer wrote: ‘Page 2, line 79: “The significance being that these degraders represent a novel therapeutic strategy to potentially treat numerous diseases.” This sentence is strange.’

Author reply: This sentence was restructured and re-written (Line 94): “These degraders, with their ability to target many protein targets have the potential to expand the druggable proteome and treat numerous diseases, highlighting their importance for future medicinal applications.”

Comment (iv): The Reviewer wrote: ‘Page 3, line 81: “Immunomodulatory agents (phthalimide derivatives: lenalidomide, thalidomide and pomalidomide) were the first ligands found to induce protein degradation.” The word “first” is incorrect. Nuclear receptor ligands have long been known to degrade their target protein degradation.’

Author reply: This sentence has been re-written to in order to correct the inaccuracy. The sentence now reads (Line 98): ‘Immunomodulatory agents (phthalimide derivatives: lenalidomide, thalidomide and pomalidomide) are a family of ligands found to induce the degradation of transcription factors IKAROS family zinc finger 1 and 3 (IKZF1/3) by interacting with cereblon, triggering their ubiquitination and degradation. [23].’

Comment (v): The Reviewer wrote:  ‘The authors should refer to two recent important papers: Nat. Chem. Biol. 2019, 15, 737–746.; Nat. Chem. Biol. 2019, 15, 747–755.’ 

Author reply: In order to correct this omission, we have included both papers into the manuscript as suggested. Each paper was discussed through the following additions made to the text:

The reference Nat. Chem. Biol. 2019, 15, 737–746 was described from line 306: ‘Recently, using small electrophilic ligands linked to known inhibitors of the predominantly nuclear proteins FKBP12 (SLF) and BRD4 (JQ1), a chemical proteomic approach identified the recruitment of DCAF6 and DTL (nuclear CUL4-DDB1 E3 ligase substrates) for targeted protein reduction [20]. The use of electrophilic ligands that can bind a broad range of targets and the subsequent identification of organelle specific E3 ligase substrates [20], highlights a possible advantage of organelle specific E3 ligase selection for specific organelle localized protein targets’.

The reference Nat. Chem. Biol. 2019, 15, 747–755 was described from line 311: ‘Nimbolide’s anti-proliferative effect against breast cancer cells was recently shown to be due to RNF114 E3 ligase binding. Therefore, it was linked to inhibitor JQ1 and used for BRD4 targeted protein degradation by recruiting RNF114 [70].’

Comment (vi): The Reviewer wrote: ‘Page 2, line 65: “This is advantageous over other modulating drugs that target specific activities without influencing other functions (such as roles in protein-protein interactions, PPIs).” This sentence is misunderstandable. If a targeted protein has roles to link two proteins, the PPI should be affected.’

Author reply: In order to clarify this statement this sentence has been re-written and now reads as (line 64): ‘Therefore, degraders have unique advantages over traditional, function specific‑targeting ligands which do not influence a targets additional functions, such as an alternative role in protein-protein interactions (PPIs) or protein scaffolding.’

Comment (vii): The Reviewer wrote: ‘Page 2, line 67: “drugs that block a specific binding site are more susceptible to drug resistance, as has been observed for several cancers…It was shown that by adding a degradation recruiting agent to enzalutamide this resistance can be overcome.” PROTACs may have the same problem and there are no guarantees they overcome.’

Author reply: The Reviewer is correct and PROTACs will likely incur resistance. However, for the purpose of this manuscript we have focused on the potential for PROTACS to overcome existing resistance observed or induced by existing molecules, and also provide examples thereof. To clarify our intention we have re-worded the sentence from line 67 as: ‘Furthermore, drugs that block a specific binding site are more susceptible to drug resistance, as has been observed for several cancers [15]. In the context of prostate cancer, the androgen receptor (AR) antagonist enzalutamide can become an agonist in the presence of a single mutation in the receptor - which is characteristic of resistance against this drug [11]. By adding a recruiter to enzalutamide, a degrader was produced with anti-proliferative effect against cell lines resistant to the drug alone [11].’

Comment (vii): The Reviewer wrote: ‘The authors should make more detailed comments on the difference between traditional PROTACs/SNIPERs and traditional ligands such as receptor antagonist/enzyme inhibitors: between reduce in protein levels and protein functional inhibition. In this point, the authors should refer to a recent review (Chem Rec 2018, 18, 1681-1700.).’

Author reply: In order to address this suggestion we expanded on the paragraph within the Introduction beginning from line 60. We also directed the readers to the excellent Review of Itoh for a more detailed comparison. The paragraph now reads: The design of bifunctional degraders with the ability to remove a protein of interest, thereby reducing its abundance, holds the potential to abrogate all of its functions. On the other hand, traditional small molecule ligands, such as enzyme inhibitors and receptor antagonists, target a specific function or activity of a protein for a modulating affect (for a detailed comparison between degraders and traditional small molecule ligands see an account by Itoh [14]). Therefore, degraders have unique advantages over traditional, function specific‑targeting ligands which do not influence a targets additional functions, such as an alternative role in protein-protein interactions (PPIs) or protein scaffolding. Furthermore, drugs that block a specific binding site are more susceptible to drug resistance, as has been observed for several cancers [15].’

Round 2

Reviewer 1 Report

The authors' updated text is a very nice improvement and I believe the text is original, comprehensive, and well integrated with the recent literature. I support publication.